# Nutrient Removal and Membrane Performance of an Algae Membrane Photobioreactor in Urban Wastewater Regeneration

**DOI:** 10.3390/membranes12100982

**Published:** 2022-10-10

**Authors:** Verónica Díaz, Laura Antiñolo, José Manuel Poyatos Capilla, Mari Carmen Almécija, María del Mar Muñío, Jaime Martín-Pascual

**Affiliations:** 1Department of Chemical Engineering, University of Granada, 18071 Granada, Spain; 2Department of Civil Engineering, University of Granada, 18071 Granada, Spain

**Keywords:** fouling, ultrafiltration membrane, microalgae, photobioreactor, wastewater reuse

## Abstract

The increase in industry and population, together with the need for wastewater reuse, makes it necessary to implement new technologies in the circular economy framework. The aim of this research was to evaluate the quality of the effluent of an algae membrane photobioreactor for the treatment of the effluent of an urban wastewater treatment plant, to characterise the ultrafiltration membranes, to study the effectiveness of a proposed cleaning protocol, and to analyse the performance of the photobioreactor. The photobioreactor operated under two days of hydraulic retention times feed with the effluent from the Los Vados wastewater treatment plant (WWTP) (Granada, Spain). The microalgae community in the photobioreactor grew according to the pseudo-second-order model. The effluent obtained could be reused for different uses of diverse quality with the removal of total nitrogen and phosphorus of 56.3% and 64.27%, respectively. The fouling of the polyvinylidene difluoride ultrafiltration membrane after 80 days of operation was slight, increasing the total membrane resistance by approximately 22%. Moreover, the higher temperature of the medium was, the lower intrinsic resistance of the membrane. A total of 100% recovery of the membrane was obtained in the two-phase cleaning protocol, with 42% and 58%, respectively.

## 1. Introduction

Microalgae are currently some of the most promising renewable raw materials to produce different subproducts [1,2,3]. From both an economic and environmental point of view, microalgae are attractive because their cultivation and processing could involve wastewater treatment and carbon dioxide capture from combustion gases [4,5,6]. Currently, microalgae-based systems are some of the most promising technologies for tertiary treatment in wastewater treatment plants (WWTPs). This is due to their associated low costs and environmental advantages [7,8].

Nowadays, microalgae cultivation has a high cost in terms of the nutrients needed. Therefore, the use of wastewater may be a viable solution because of its nutrient content, coupled with the ability of these micro-organisms to tolerate very wide conditions of temperature, pH, or salinity [9,10]. Wastewater provides the nutrients necessary for the growth of the microalgae, in turn reducing the cost of production and greenhouse gas emissions because the microalgae will absorb some of the carbon generated in the wastewater treatment that would be released as carbon dioxide [6,9,11]. This achieves the requirements of wastewater discharge standards, which specify very low nutrient concentrations to avoid eutrophication [12].

Wastewater treatment with microalgae reduces the costs of microalgae biomass production by 40% of the total cost, generating a lower energy demand for nutrient removal and, therefore, presenting lower costs compared to conventional wastewater treatment [13,14,15]. On one hand, microalgae can use inorganic phosphorus and nitrogen for their growth, as well as produce oxygen during photosynthesis. On the other hand, microalgae are also used for the removal of coliforms because the environmental growth conditions for microalgae are unfavourable for these microorganisms [16]. Both phosphorus and nitrogen can be considered limiting factors in the growth of microalgae [17].

Much of the literature on the use of microalgae for wastewater treatment focuses on the treatment of synthetic wastewater using pure microalgae strains, ignoring bacteria and other microorganisms present in real wastewater. For this reason, reproducibility from laboratory to real WWTP conditions is complicated. In WWTPs, a heterogeneous and complex consortia of microalgae, cyanobacteria, and other microorganisms would be used [18]. Therefore, we decided to work with real treated urban wastewater in this study.

It is known that biological wastewater treatment with activated sludge is energy-intensive. Therefore, the use of bacterial and microalgae consortia could be an alternative treatment method [19]. In this way, aeration costs could be saved due to the algae’s ability to produce sufficient oxygen for bacterial growth and to adsorb ammonium and phosphate [20], just as the bacteria provide carbon dioxide for the growth of the microalgae [21]. Some of the microalgae used for wastewater treatment belong to the genera *Scenedesmus* sp., *Chlorella* sp., *Desmodesmus* sp., *Neochloris* sp., *Chlamydomonas* spp., and *Nitzschia* spp. [7,22]. 

One of the most promising technologies widely used for municipal and industrial wastewater treatment is the membrane photobioreactor (MPBR), due to its advantages in terms of small footprint, high quality of treated water, and easy operation [23,24].

Membranes are a very promising technology for harvesting microalgae cultures [25]. This is due to the energy and cost savings compared to using centrifuges, as well as the fact that they retain almost all of the biomass [12]. According to other studies, the microalgae concentration in a photobioreactor without a membrane is significantly lower than in an MPBR [12,26]. In addition, membrane filtration in the photobioreactor prevents the microalgae culture from being washed out, as the solid retention time (SRT) in the photobioreactor will be different from the hydraulic retention time (HRT). In this way, higher yields and biomass concentrations are achieved [12,27,28]. This also affects the concentration of nutrients, especially when working with domestic wastewater, in which nitrogen and phosphorus concentrations are relatively low [29]. Another advantage of using membranes in photobioreactors is the reduction in volume, as the removal of nutrients present in urban wastewater would require less space [12]. However, a potential drawback of MPBRs is that if the effluent is untreated wastewater, it can lead to the death of the microalgae species being cultivated in the effluent, in which case it is necessary to design an appropriate pre-treatment. In the same way, it is important to correctly select the species of microalgae to be cultivated, as not all of them are capable of adapting to the conditions of the wastewater [30]. Another disadvantage of these photobioreactors is the high risk of contamination of the microalgae culture, as well as the tedious and costly work involved in harvesting the microalgae [31]. 

Previous authors have claimed that the maintenance and operating costs of an MPBR are almost eight times lower than those of conventional nutrient removal treatments [32]. Even the energy costs associated with an MPBR would be lower than those used in closed photobioreactors or open ponds [30]. According to other authors, the potential of MPBRs for domestic wastewater treatment using a consortium of bacteria and microalgae has not yet been studied [20]. 

In the present investigation, the continuous performance of a photobioreactor equipped with a polyvinylidene fluoride (PVDF) ultrafiltration membrane for the treatment of urban wastewater already treated with microalgae with a two-day HRT was studied to achieve the following objectives: (1) evaluation of the water quality of the effluent obtained in a photobioreactor equipped with an ultrafiltration membrane for the treatment of already treated urban wastewater from a WWTP using microalgae culture, (2) analysis of the performance of the photobioreactor, and (3) evaluation of the operation of the ultrafiltration membrane in a microalgae culture. 

## 2. Materials and Methods

### 2.1. Pilot Plant

This study was carried out in a cylindrical methacrylate photobioreactor of 18 L (Figure 1). It was equipped with a hollow-fibre polymeric ultrafiltration membrane (PVDF) configured according to the needs of the plant, and manufactured from a reconditioned membrane with a larger unit surface area (0.92 m^2^), a nominal pore size of 0.04 μm, and a membrane life of approximately 10 years.

The fibres of the ultrafiltration membrane were taken and divided into four smaller filtration modules, adapted to the photobioreactor used in this research. To this end, the fibres that make up the new module were sealed with silicone to the PVC parts that allow the membrane to be connected to the filtration system. The length of each hollow fibre was 500 mm, and the total membrane surface area of this new membrane was 0.2 m^2^. 

The permeate was extracted through the membrane by a suction-backwashing peristaltic pump (323 U, Watson-Marlow Pumps Group, Marlow, UK). The permeate extraction and backwashing periods were 9 and 1 min, respectively.

To automate and control the correct operation of the photobioreactor, a vacuum and pressure gauge (Potermic, Barcelona, Spain) were installed to control the pressure during the filtering process and level probes to keep the photobioreactor at an approximately constant volume. In addition, the photobioreactors have a continuous stirring system of approximately 8 Hz (MSL 63 A, Cime Motors, Girona, Spain) and continuous air injection by means of a circular diffuser at the base of the photobioreactor. In addition to natural lighting, the photobioreactor was illuminated by fluorescent lights (L36 W/868, OSRAM DS, Munich, Germany) placed around the plant, providing an average irradiance in the photobioreactor of 13.24 µmol/m^2^·s. A 12/12-h light/dark cycle was established by means of a timer.

To determine the average irradiance (light intensity) in the photobioreactor, an HD 2303.2 Delta OHM light meter (Arava, Spain) was used with the DLP471 PAR probe (Arava, Spain) for photon measurement. Measurements were taken on and around the photobioreactor surface, both perpendicular and parallel to the photobioreactor.

The photobioreactor was continuously fed with treated urban wastewater from real wastewater taken from WWTP of Los Vados (Granada, Spain), and naturally occurring microalgae species were cultured. The HRT set in the photobioreactor was two days, where the transmembrane operating pressure (TMP) was approximately 0.01 bar. The experimental cycle lasted 80 days, coinciding with the SRT.

### 2.2. Experimental Procedure

Samples were taken on alternate days in the influent, photobioreactor, and effluent. Once the samples were collected, the pH was measured with a Crison pH 25 meter (Hach, Barcelona, Spain), and the temperature and conductivity were measured using the multifunctional meter PCE-PHD 1 (PCE Ibérica SL, Alicante, Spain). For the analysis of Total Suspended Solids (TSS) present in the sample, 0.45 µm Millipore glass fibre filters were used following standard methods [33]. The filtrate was used for the analysis and quantification of nutrients in the samples, determined by ion chromatography using a conductivity detector (Metrohm, Metrohm AG, Herisau, Switzerland). Finally, the Heλios spectrophotometer (Thermo Spectronic, Madrid, Spain) was used to measure turbidity at a wavelength of 650 nm and the optical density of the culture at 680 nm, following the same procedure as other authors [8,34].

### 2.3. Kinetic Modelling

Table 1 shows the six kinetic models that were evaluated to adjust the kinetics of microalgae biomass growth in the photobioreactor, measured as TSS throughout the evolution of the concentration over the time (*C_t_*): 

$C_{0}$ (g·L^−1^) represents the initial biomass concentration, $C_{t}$ (g·L^−1^) is the biomass concentration at any time, and $C_{e}$ (g·L^−1^) is the biomass equilibrium concentration. The experimental time is expressed as t (days) and the model constant as k (days^−1^). The parameter h relates the biomass concentration at equilibrium and the kinetic model constant as follows: $h=k\cdot C_{e}^{2}$ (g^2^·L^−2^·days^−1^).

The data were fitted to the models described above, minimising the sum of squared errors between the empirical and modelled data using the Microsoft Excel Solver application.

### 2.4. Membrane Filtration Tests

The membrane was characterised prior to use to determine the intrinsic resistance of the clean membrane. For this purpose, a test was carried out with clean water. Subsequently, the behaviour of the clean membrane was studied with the content of the photobioreactor (microalgae culture in treated urban wastewater).

At the end of the study cycle, the membrane was again characterised with water and the content of the photobioreactor. In this way, the total resistance of the membrane after fouling was obtained.

Finally, the recovery of the membrane permeability was studied by means of a two-stage cleaning protocol. In the first stage of the cleaning protocol, the membrane was inorganically cleaned with citric acid (800 ppm) for 5 h. The second cleaning step consisted of an organic cleaning with sodium hypochlorite (800 ppm) for 5 h. After each of the cleaning stages, the membrane was characterised in water to evaluate the cleaning efficiency of each stage.

Each of the tests was carried out at three different temperatures (20, 25, and 30 °C). To carry them out, the volumes obtained after 30 s of filtration at different values of TMP were recorded. In this way, filtration fluxes were obtained for each of the TMP sets.

#### 2.4.1. Membrane Characterisation

The membrane permeability (*K*) was calculated from the following equation as described above [35], where *J* is the permeate flux (LHM, litres per square metre per hour filtered) and *TMP* is the measured transmembrane pressure (bar): (1)$$J=K\cdot TMP=\frac{TMP}{µ\left(R_{M}+R_{CP}+R_{F}\right)}=\frac{TMP}{R_{M}^{\prime }+R_{F}^{\prime }}$$

*R_M_* represents the intrinsic resistance of the membrane, *R_CP_* is the resistance that occurs in the first moments of filtration, which is reversible, and *R_F_* corresponds to the resistance that occurs at longer times, which corresponds to fouling. *R_M_*′ and *R_F_*′ represent the total value of the intrinsic membrane resistance and the total value of the fouling resistance, respectively, by integrating the viscosity constant for each temperature.

This permeability constant encompasses the influence of temperature and the intrinsic resistance of the membrane. On the other hand, the intrinsic membrane resistance (*R_M_*′) was calculated as the inverse of the permeability, which had been obtained with the membrane cleaned in water:(2)$${R_{M}}^{\prime }=\frac{1}{K}$$

The fouling resistance (*R_F_*′) of the membrane was calculated as described in Equation (3), where *R_i_*′ represents the total membrane resistance in each of the cases studied.
(3)$${R_{F}}^{\prime }={R_{i}}^{\prime }-{R_{M}}^{\prime }$$

#### 2.4.2. Recovery of Membrane Characteristics

The efficiency of the membrane cleaning was calculated as shown in Equation (4), where *R*_0_ represents the total membrane resistance with a soiled membrane, *R_i_*′ represents the total membrane resistance in each of the cases studied, *R*′_*i*−1_ represents the total membrane resistance of the case before the one under study, and *R_M_*′ is the intrinsic membrane resistance:(4)$$E\;\left(\%\right)=\frac{R_{i-1}^{\prime }-{R_{i}}^{\prime }}{R_{0}-{R_{M}}^{\prime }}\cdot 100$$

#### 2.4.3. Analysis of the Operational TMP

The TMP was determined by setting the flow rate necessary for the HRT in the photobioreactor of the treated urban wastewater to be two days. It was verified that the operating TMP of the membrane was in the pressure-controlled working region.

To evaluate the flux at different TMPs using the contents of the photobioreactor, tests were carried out with both clean and dirty membranes after the end of the experimental cycle in the photobioreactor. For this purpose, the equation of the series resistance model described below was used:(5)$$J=\frac{TMP}{R_{M}^{\prime }+b\cdot TMP}$$

The parameter *b* is a constant which relates the behaviour of the TMP to the matter transfer in the filtration process.

## 3. Results and Discussion

### 3.1. Photobioreactor Operation

After analyses of the samples were taken, the average operating conditions of the reactor during the experimental period were 7.40 ± 0.88 for pH, 15.54 ± 0.89 °C for temperature, and 954.30 ± 75.82 µS/cm for conductivity.

Figure 2a shows the evolution of the TSS in the photobioreactor during the experimental time. The concentration of algal biomass increases as time passes until the stationary phase is reached. In this case the results obtained during the growth phase have been represented up to their maximum point, where the stationary phase, not represented, would begin.

In the same way, the turbidity results obtained in the photobioreactor are shown in Figure 2b and expressed in NTU (Nephelometric turbidity unit), which are related to the TSS concentration in the photobioreactor, increasing as the concentration of microalgae biomass increases.

Figure 3 shows the evolution of the optical density of the microalgae culture. These values were obtained by taking the difference between the absorbance measurement at 680 nm of the microalgae photobioreactor culture and the measurement obtained for the microalgae feed (treated urban wastewater).

The same growth trend was observed in the three parameters studied, with a difference—in the study of the optical density, the interferences that could be caused by the TSS contained in the treated urban wastewater, bacteria, and other microorganisms that influence the correct evaluation of the growth of the algal biomass were eliminated. Figure 4 shows the microalgae present in the photobioreactor, which grew naturally in the treated urban wastewater.

The kinetic constants and correlation rates obtained for each of the kinetic models evaluated for the concentration of TSS and for the optical density obtained in the photobioreactor, which has been evaluated at a wavelength of 680 nm, are presented in Table 2:

Comparing the results obtained, in the case of the TSS, the kinetic model with the highest correlation index is the pseudo-second-order model. However, in the case of optical density, the best correlation rate is obtained for the zero-order model. Figure 5 and Figure 6 show graphical representations of the adjustments that were made for the TSS and optical density (OD) data.

The kinetic constants obtained for the fit of the optical density data show, in absolute value, had higher values compared to those obtained for the TSS, except for the pseudo-second-order model. This indicates that the overall growth observed is slower than the growth of micro-organisms detected at 680 nm, including microalgae.

The best correlation rate, and therefore the best fit, was obtained for the zero-order model when evaluating the optical density values. These results are considered satisfactory because the optical density measurement eliminated interferences caused by bacteria or other microorganisms in the TSS measurement.

The kinetic growth of microalgae can be characterised by a phase in which the specific growth rate exists until it reaches a maximum value, where it is maintained for a specific time. This kinetic growth model corresponds to the one described by Monod, one of the most widely used [7,36]. No substrate-dependent data are available to fit this model, as the photobioreactor is assumed to operate under perfect mixing conditions. Therefore, the nutrient concentrations (the substrate) are approximately constant throughout the experiment. For this reason, the fit of the experimental data to the Monod growth kinetic model could not be determined.

No studies evaluating microalgae growth using polynomial kinetic models have been found in the literature. However, they have been evaluated to represent the growth of microalgae attached to support materials [37,38,39].

### 3.2. Membrane Operation 

#### 3.2.1. Membrane Characterisation

As described above, before starting the study in the photobioreactor, the resistance of the cleaned membrane at different temperatures was characterised.

As expected, as the temperature increases, the membrane resistance decreases, taking values between 6.21 × 10^−3^ and 5.8 × 10^−3^ bar·m^2^·h/L (Table 3). As previously studied by other authors, increasing the temperature increases the permeability of PVDF ultrafiltration membranes, or in other words, the membrane resistance decreases, as shown in our results [35,40].

Once the photobioreactor study was completed after 80 days of filtration, the membrane was characterised to assess its fouling. The data obtained are shown in Table 4, with the total membrane resistances (*R_i_*′) and the fouling resistances (*R_F_*′) for each of the temperatures tested. For this purpose, the permeate flux and transmembrane pressure results were processed to obtain the membrane permeability (*k*) in each case. Knowing the relationship of this parameter with the membrane resistance (Equation (2)), *R_i_*′ was obtained for each of the tests carried out. The *R_M_*′ was obtained in the same way using the data of the clean membrane. Subsequently, and taking into account Equation (3), the *R_F_*′ was obtained in each case. The correlation rate corresponds to the data fit by forcing the line through the origin.

The *R_F_*′ values should be approximately the same for all temperatures, since the fouling that has occurred on the membrane is the same for all three tests. These variations are due to the small membrane washout that has occurred over time during the three characterisation tests, as well as the effect of temperature, because it affects the viscosity of the filtered fluid.

As in the case of intrinsic resistance, the total resistance of the membrane decreases as the temperature increases. The fouling resistance values obtained were between 1.67 × 10^−3^ and 1.01 × 10^−3^ bar·m^2^·h/L; the membrane resistance increased due to fouling by approximately 22%. Therefore, the fouling produced by the contents of the photobioreactor (including microalgae) is small, as has been found in previous studies [27].

Low fouling ensures that the lifetime of the ultrafiltration membrane will be longer because the permeability of the membrane is not significantly affected and therefore the costs associated with pumping do not increase [41].

#### 3.2.2. Evaluation of the Effectiveness of the Cleaning Protocol and Recovery of Membrane Characteristics

To recover the filtering characteristics of the membrane, the cleaning protocol described above was carried out. After each cleaning step, the membrane was characterised in water to calculate the efficiency of each cleaning step (Table 5) according to Equation (4).

In all cases, the initial resistance of the membrane was recovered, and the overall effectiveness of the cleaning protocol was 100%. As can be seen from the results, in the first cleaning step, a recovery of approximately 42% was obtained. On the other hand, in the second cleaning step, the remaining 58% was recovered, reaching the initial *R_M_*′ values.

Both steps were effective and therefore necessary for complete cleaning of the fouled membrane in the photobioreactor characteristics. In the tests carried out at different temperatures, similar results were obtained, with variation in the values of the membrane resistance because of temperature. Therefore, the optimal temperature for the cleaning protocol is considered to be 30 °C, where a lower *R_M_*′ value is obtained.

This protocol would be cost-effective at an industrial level because it is a discontinuous cleaning process which can be sized according to the membranes in large installations to minimise costs, limiting and spacing the cleaning to four times per year. This also highlights the usefulness of this cleaning protocol, which can be used on a small scale with very good results. Initially, only organic cleaning with sodium hypochlorite could be carried out, minimising costs, since from the trials it is expected to be at least 50% effective. Previous authors applied another cleaning protocol after the use of the membranes in a microalgae culture. Their protocol was less effective than the one applied in our study because the inorganic fouling was not successfully removed, so they suggested the use of citric acid to complete the cleaning, and this has been done in our study [27].

#### 3.2.3. Analysis of the Operational TMP

To determine the optimal working TMP for our study system, fluxes (J) at different TMPs were analysed to fit the data to the Equation (5) model. In no case was it possible to fit the proposed equation, because the studied TMPs allowed by the filtration device were in the pressure-controlled zone, without reaching the values of the zone controlled by matter transfer. Therefore, it was decided to fit these data to the linear part of the equation.

Figure 7 shows the data obtained for the three temperatures studied with the membrane clean before starting the study cycle and after finishing it (with the membrane soiled). From these representations, the data shown in Table 6 were obtained:

The fouling that occurs is so small that the *R_M_*′ values obtained with water and with the photobioreactor contents are similar both at the beginning and at the end of the cycle. The low fouling obtained in the membrane may be due to the consortium of bacteria and algae that exist in the MPBR, decreasing the organic load present in the medium and therefore decreasing fouling [42]. Again, the behaviour is similar in the three studies at different temperatures, obtaining a variation only due to the influence of temperature. Membrane resistance increases between 28 and 13% with temperature.

Again, the resistance values obtained for the fouled membrane should be approximately the same for all temperatures, as the fouling that has occurred on the membrane is the same for all three tests. These variations are due to the small amount of membrane washout that has occurred over time during the three characterisation tests. As before, these differences are observed because of the effect of temperature on the viscosity of the filtered fluid.

As can be seen, the selected working TMP, which is 0.01 bar, is in the linear zone and therefore within the recommended range for the operation of this system, being in the pressure-controlled zone. This TMP is similar to that used in other studies in which photobioreactors with microalgae are used [20]. Working under these TMP conditions favours less fouling of the membrane. This effect is explained by the fact that at higher TMP values, greater membrane resistance is caused by the compression of the layer of cells deposited on it [43].

### 3.3. Effluent Quality

The quality of the effluent from the photobioreactor was assessed by analysing the TSS, turbidity, and quantification of nitrogen and phosphorus. Table 7 shows the average values of temperature, pH, conductivity, TSS, and the turbidities of influent and effluent.

According to Regulation (EU) 2020/741 of 25 May 2020, which establishes minimum requirements for water reuse in agricultural irrigation, the effluent from the photobioreactor can be reused as reclaimed water quality class A, B, C, and D [44]. In other words, it complies with all the minimum requirements set out in that regulation.

According to Royal Decree 1620/2007, of 7 December 2007, which establishes the legal regime for the reuse of treated water and in relation to the results obtained for the concentration of TSS and turbidity, the effluent from our photobioreactor could be reused for urban uses of quality 1.1 and 1.2, agricultural uses of quality 2.1, 2.2, and 2.3, industrial uses of quality 3.1, recreational uses of quality 4.1 and 4.2, and environmental uses of quality 5.1, 5.2, and 5.3 [45]. Therefore, the effluent obtained satisfies the requirements established for the reuse of treated wastewater in all of the uses regulated in this Royal Decree, except for quality 3.2 (industrial uses in cooling towers and evaporative condensers).

#### Nitrogen and Phosphorus Removal

Table 8 shows the results obtained in the nutrient analysis: the average phosphorus and nitrogen in the influent and effluent, the removal yields, and the consumption of each nutrient. These last two parameters were calculated considering that the HRT in the photobioreactor is 2 days.

The total nitrogen removal performance was slightly lower than that obtained by other authors [7]. This effect is probably due to the presence of autotrophic microorganisms in the wastewater. These consume the ammonium (reducing it in the photobioreactor) but generate nitrates, increasing their concentration [20]. However, the total nitrogen removal rate in the photobioreactor is higher than that obtained by other authors [46].

Other authors report that working times at high SRTs influence the feeding preference of microalgae, observing that for long SRTs ammonium is preferred to nitrate, raising nitrate levels in the effluent [28]. The preference for nitrate or ammonium will depend on the species of microalgae grown in the photobioreactor and its operational conditions [7].

One of the possible solutions to improve the nutrient yield in the photobioreactor would be to increase the HRT. In this way, the dilution of the culture is not so great, and therefore, the renewal of the substrate is not so fast, thus improving the elimination yields, as the microalgae have more time to assimilate the nutrients [12].

On the other hand, it has been observed that phosphorus concentrations have been quite low in the influent of the photobioreactor. This event has probably conditioned the growth of the microalgae, acting as a limiting nutrient for the culture. To achieve higher nutrient removal yields under the same operating conditions, a higher amount of biomass would be necessary, and this would require a higher concentration of phosphorus in the medium. Despite this, the average removal efficiency obtained for phosphorus is higher than that obtained in other studies [25]. Low phosphate levels cannot be ignored as this effect can reduce nitrate uptake, limiting microalgae growth and nutrient removal performance in the photobioreactor [7,47].

## 4. Conclusions

The following conclusions were reached after 80 days under continuous operation (two days of HRT) of a microalgae membrane photobioreactor for the treatment of real urban wastewater effluent by culturing the microalgae present in it.

On one hand, by analyzing the results obtained, it was observed that optical density was the parameter that best described the growth kinetics of microalgae present in the photobioreactor. The zero-order model had the highest correlation rate of 0.9776, and a value of 8.921 × 10^−3^ days^−1^ was obtained for the kinetic constant k. 

On the other hand, with respect to the membrane, there was an increase in the total membrane resistance of approximately 22%. Therefore, under the operative conditions tested, the fouling was low. A two-phase cleaning protocol (inorganic and organic cleaning) was proposed to recover membrane permeability. The recoveries were approximately 42 and 58% in the inorganic and organic phases, respectively, for the temperatures tested (20, 25, and 30 °C), but *R_M_*′ values were lower as the temperature increased, as would be expected. At all temperatures, the overall effectiveness of the cleaning protocol was 100%. However, the optimum temperature tested for the cleaning protocol is considered to be 30 °C, where a lower *R_M_*′ value is obtained. 

Regarding the effluent obtained and the removal of the nutrients present in the treated wastewater, total nitrogen and phosphorus removals in the algae membrane photobioreactor were 56.3 and 64.27%, respectively. Therefore, taking into account other water quality parameters, the effluent of the microalgae membrane photobioreactor could be reused for agricultural irrigation, according to Regulation (EU) 2020/741 of 25 May 2020.

Considering the above, a microalgae membrane photobioreactor could be an appropriate technology for urban wastewater regeneration to be included in a conventional activated sludge treatment plant.

## Figures and Tables

**Figure 1 membranes-12-00982-f001:**
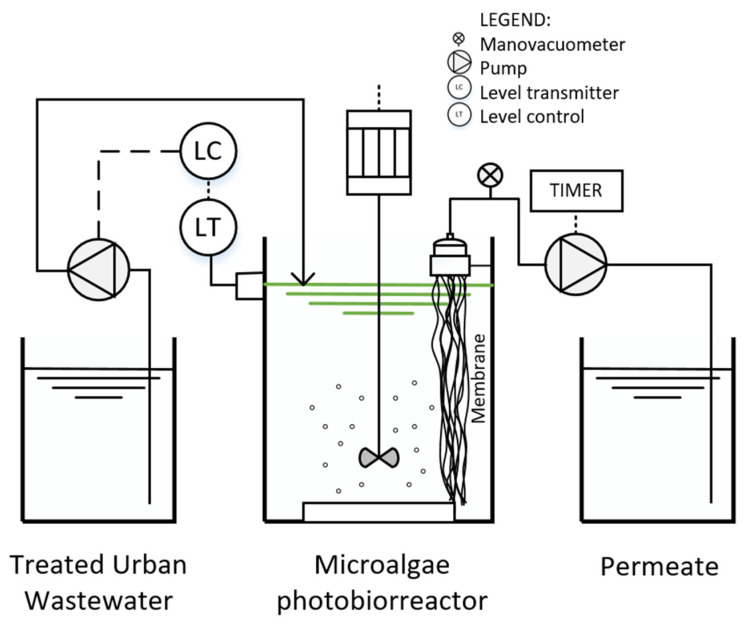
Experimental pilot plant used in the laboratory.

**Figure 2 membranes-12-00982-f002:**
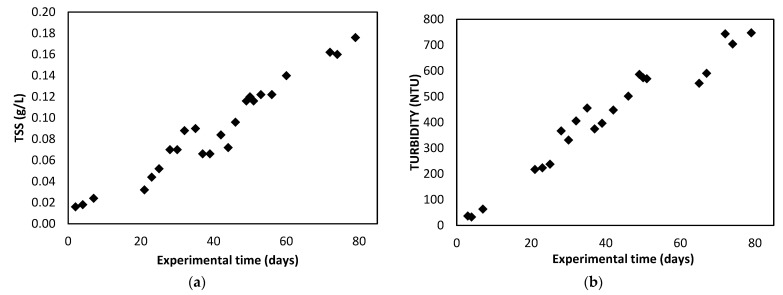
Evolution of TSS (**a**) and turbidity (**b**) in the photobioreactor during the experimental period.

**Figure 3 membranes-12-00982-f003:**
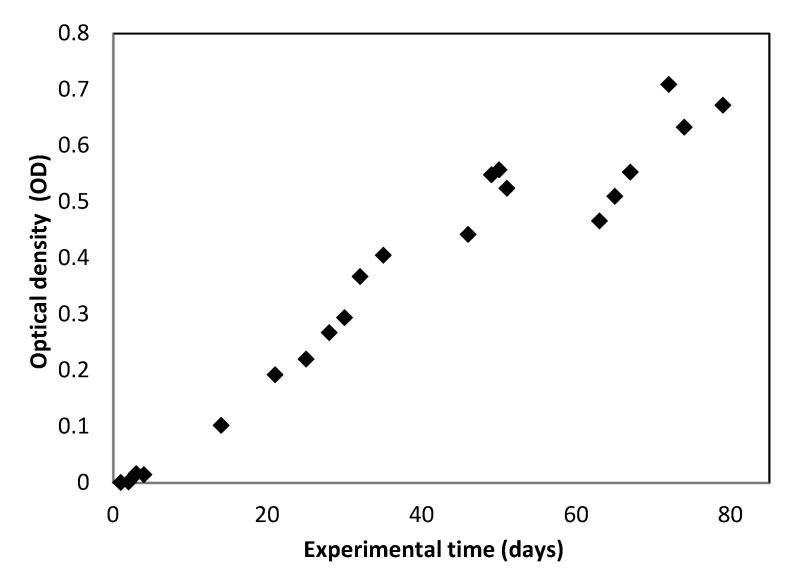
Evolution of optical density (OD) in the photobioreactor during the experimental period.

**Figure 4 membranes-12-00982-f004:**
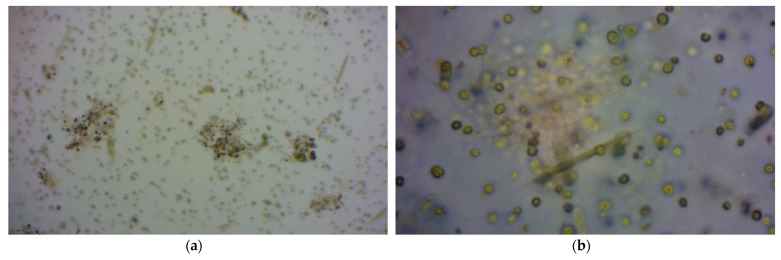
(**a**) Microscope photograph of photobioreactor contents. (**b**) Enlarged microscope photograph of microalgae culture.

**Figure 5 membranes-12-00982-f005:**
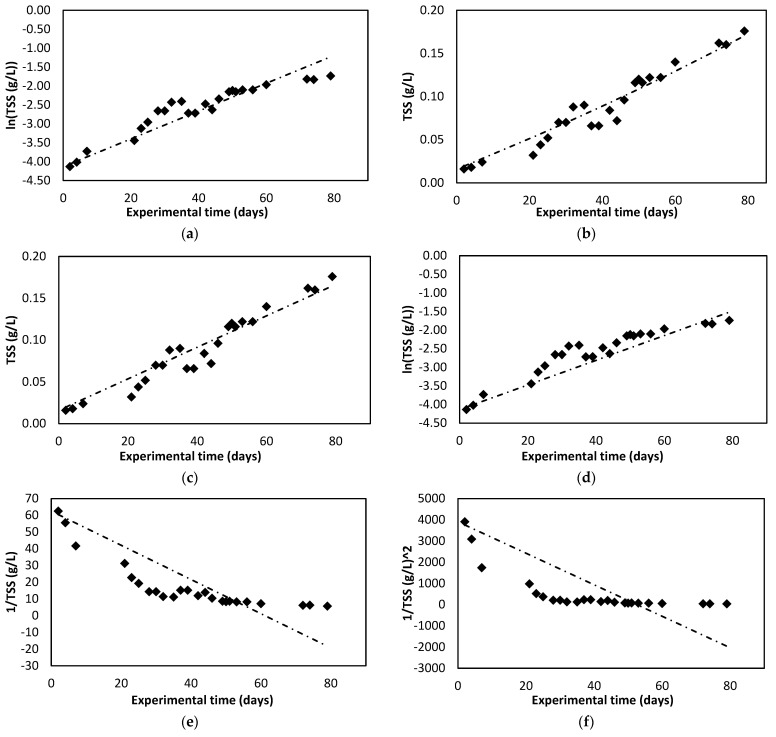
Fitting data for TSS to each kinetic model assessed. (**a**) Pseudo-first order. (**b**) Pseudo-second order. (**c**) Zero order. (**d**) First-order. (**e**) Second-order. (**f**) Third order.

**Figure 6 membranes-12-00982-f006:**
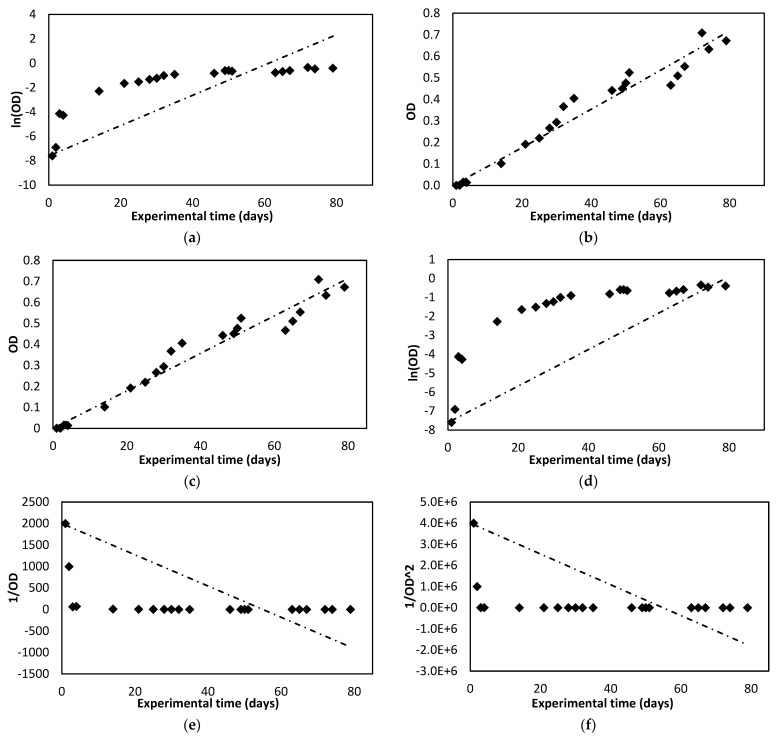
Fitting data for optical density (OD) (absorbance at 680 nm) to each kinetic model assessed. (**a**) Pseudo-first order. (**b**) Pseudo-second-order. (**c**) Zero order. (**d**) First-order. (**e**) Second-order. (**f**) Third order.

**Figure 7 membranes-12-00982-f007:**
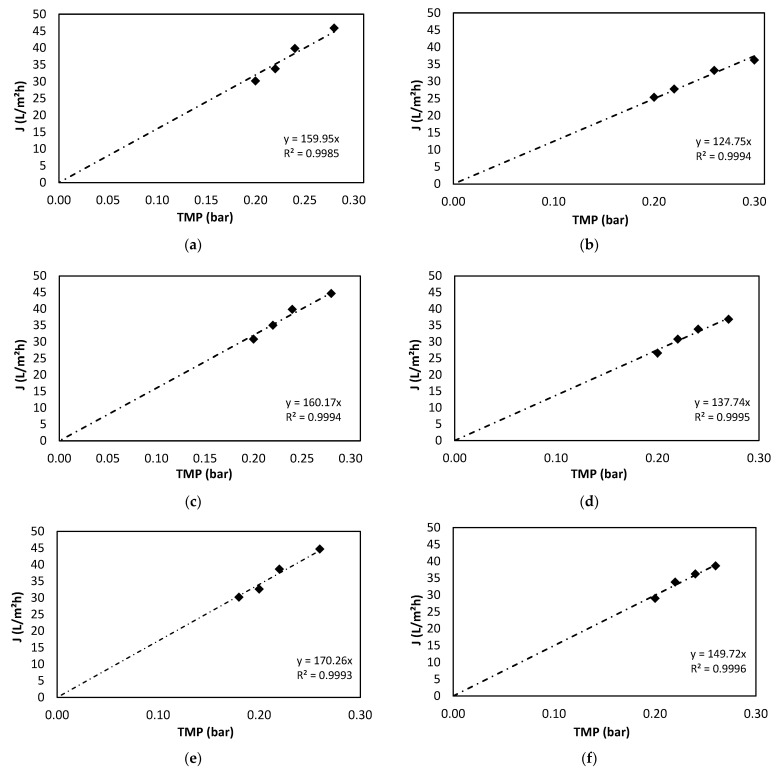
Fitting the permeate flux (J) and TMP data at different temperatures to the linear part of the per-formance model. (**a**) Clean membrane at 20 °C. (**b**) Soiled membrane at 20 °C. (**c**) Clean mem-brane at 25 °C. (**d**) Soiled membrane at 25 °C. (**e**) Clean membrane at 30 °C. (**f**) Soiled membrane at 30 °C.

**Table 1 membranes-12-00982-t001:** Kinetic models.

Kinetic Model	Equation
Pseudo-first-order	$lnC_{t}=lnC_{0}-kt$
Pseudo-second-order	$C_{t}=C_{0}-\frac{t}{\left(\frac{1}{h}+\frac{t}{C_{e}}\right)}$
Zero-order	$C_{t}=C_{0}+k_{0}t$
First-order	$\ln\left(C_{t}\right)=lnC_{0}+k_{1}t$
Second-order	$\frac{1}{C_{t}}=\frac{1}{C_{0}}+k_{2}t$
Third-order	$\frac{1}{C_{t}^{2}}=\frac{1}{C_{0}^{2}}+k_{3}t$

**Table 2 membranes-12-00982-t002:** Kinetic constants and correlation rates for each evaluated kinetic model.

Kinetic Model	Total Suspended Solids (TSS)		Optical Density (OD)	
	Model Constants	Correlation Rate R^2^	Model Constants	Correlation Rate R^2^
Pseudo-first order	$k=-3.677\cdot 10^{-2}\;\mathrm{day}s^{-1}$	0.9444	$k=-1.243\cdot 10^{-1}\;\mathrm{day}s^{-1}$	0.7897
Pseudo-second order	$C_{e}=1.075\;g/L$ $k=-1.499\cdot 10^{-3}\;\mathrm{day}s^{-1}$	0.9735	$C_{e}=2.316\;g/L$ $k=-1.623\cdot 10^{-5}\;\mathrm{day}s^{-1}$	0.9764
Zero order	$k_{0}=1.881\cdot 10^{-3}\;\mathrm{day}s^{-1}$	0.9715	$k_{0}=8.921\cdot 10^{-3}\;\mathrm{day}s^{-1}$	0.9776
First-order	$k_{1}=3.312\cdot 10^{-2}\;\mathrm{day}s^{-1}$	0.9444	$k_{1}=9.625\cdot 10^{-2}\;\mathrm{day}s^{-1}$	0.7897
Second-order	$k_{2}=-1.023\;\mathrm{day}s^{-1}$	0.8349	$k_{2}=-36.367\;\mathrm{day}s^{-1}$	0.4789
Third order	$k_{3}=-7.431\cdot 10^{1}\;\mathrm{day}s^{-11}$	0.7264	$k_{3}=-7.288\cdot 10^{4}\;\mathrm{day}s^{-1}$	0.4118

**Table 3 membranes-12-00982-t003:** Intrinsic resistance values of the ultrafiltration membrane at different temperatures.

Temperature (°C)	*R_M_′* Values (bar·m^2^·h/L)	Correlation Rate R^2^
20	$6.21\times 10^{-3}$	0.9997
25	$5.90\times 10^{-3}$	0.9994
30	$5.80\times 10^{-3}$	0.9979

**Table 4 membranes-12-00982-t004:** Total resistance and fouling resistance of the membrane at different temperatures after fouling.

Temperature (°C)	*R_i_′* Values (bar·m^2^·h/L)	*R_F_*′ Values (bar·m^2^·h/L)	Correlation Rate R^2^
20	$7.88\times 10^{-3}$	$1.67\times 10^{-3}$	0.9997
25	$7.18\times 10^{-3}$	$1.28\times 10^{-3}$	0.9989
30	$6.80\times 10^{-3}$	$1.01\times 10^{-3}$	0.9999

**Table 5 membranes-12-00982-t005:** Results obtained for the different phases of the cleaning protocol at each of the set temperatures.

Cleaning Step	Temperature (°C)	*R_i_*′ Values (bar·m^2^·h/L)	Efficiency (%)	Correlation Rate R^2^
1	20	$7.12\times 10^{-3}$	45.13%	0.9997
	25	$6.64\times 10^{-3}$	42.13%	0.9963
	30	$6.38\times 10^{-3}$	42.01%	0.9995
2	20	$6.21\times 10^{-3}$	54.87%	0.9997
	25	$5.90\times 10^{-3}$	57.87%	0.9994
	30	$5.80\times 10^{-3}$	57.99%	0.9979

**Table 6 membranes-12-00982-t006:** Clean and soiled membrane resistances at different temperatures.

Temperature (°C)	Clean Membrane		Soiled Membrane	
	*R_M_*′ Values (bar·m^2^·h/L)	Correlation Rate R^2^	*R_i_*′ Values (bar·m^2^·h/L)	Correlation Rate R^2^
20	$6.25\times 10^{-3}$	0.9985	$8.02\times 10^{-3}$	0.9994
25	$6.24\times 10^{-3}$	0.9994	$7.26\times 10^{-3}$	0.9995
30	$5.87\times 10^{-3}$	0.9993	$6.68\times 10^{-3}$	0.9996

**Table 7 membranes-12-00982-t007:** Average values obtained for the parameters analysed in the influent and effluent.

Sample	pH	Temperature (°C)	Conductivity (µS/cm)
Influent	7.71 ± 0.26	15.30 ± 0.96	1153.90 ± 95.71
Effluent	7.41 ± 0.57	14.97 ± 0.80	970.36 ± 67.10

**Table 8 membranes-12-00982-t008:** Results of nutrient analysis.

Sample	Phosphorus	Nitrogen
Influent	2.01 ± 1.15 ppm	36.54 ± 13.93 ppm
Removal yields	64.27 ± 0.29%	56.30 ± 0.13%
Nutrient consumption	1.09 ± 0.66 ppm	20.98 ± 9.54 ppm

## Data Availability

The data presented in this study are available on request from the corresponding author.

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
