# Peer review of "Nutrient Removal and Membrane Performance of an Algae Membrane Photobioreactor in Urban Wastewater Regeneration"

_membranes, 2022, doi:10.3390/membranes12100982_

Round 1

Reviewer 1 Report

Introduction may be condensed.

The terms ‘Reactor’ and ‘Photobioreactor’ are used interchangeably or both are used in different context. This may be looked into.

It is stated that permeability constant is influenced by temperature. However, under conclusion it is stated that 42% and 58% recoveries are independent of temperature. This may be checked.

Optimum recovery is at what temperature condition?

Membrane life may be indicated.

Cost-effectiveness of the proposed method should be highlighted.

Usefulness of method for regular use may also be indicated, if it is so.    

Author Response

Thank you very much for your revision and your considerations about the paper. We hope the content of this manuscript has been improved after the revision with your comments. We will answer each of them.

Introduction may be condensed.

Answer: Thank you so much for your recommendation. Considering this suggestion and those of the other reviewers, an attempt has been made to reduce the content of the introduction in the manuscript.

The terms ‘Reactor’ and ‘Photobioreactor’ are used interchangeably or both are used in different context. This may be looked into.

Answer: Thank you so much for your consideration. The document has been revised and the terms "reactor" and "photobioreactor" have been changed to facilitate understanding.

It is stated that permeability constant is influenced by temperature. However, under conclusion it is stated that 42% and 58% recoveries are independent of temperature. This may be checked.

Answer: Thank you so much for your recommendation. This error has been revised and corrected, because, as it indicates, the membrane recoveries depend on the temperature, although in our tests we obtained approximate values for each of them. The sentence has been modified as follows: “The recoveries were approximately 42 and 58% in the inorganic and organic phases, respectively, for the temperatures tested (20, 25 y 30°C), but with lower RM’ values as the temperature increased, as would be expected.”

Optimum recovery is at what temperature condition?

Answer: Thank you so much for your consideration. Paragraph 3.2.2 and the conclusions have been revised to include the assessment of the optimum temperature for carrying out the cleaning protocol. The following sentence has been included in the text: “At all temperatures, the overall effectiveness of the cleaning protocol was 100%, however, the optimum temperature tested for the cleaning protocol is considered to be 30ºC, where a lower RM’ value is obtained.”

Membrane life may be indicated.

Answer: Thank you so much for your consideration. Membrane life (approximately 10 years) has been indicated in the section on materials and methods.

Cost-effectiveness of the proposed method should be highlighted.

Answer: Thank you so much for your suggestion. The following text has been included in the manuscript: “This protocol would be cost-effective at an industrial level because it is a discontinuous cleaning process which can be sized according to the membranes in large installations to minimise costs. Limiting and spacing the cleaning to 4 times per year.”

Usefulness of method for regular use may also be indicated, if it is so.    

Answer: Thank you so much for your consideration. The following text has been included in the manuscript: “It also highlights the usefulness of this cleaning protocol which can be used on a small scale with very good results. Initially, only organic cleaning with sodium hypochlorite could be carried out, minimising costs, since from the trials it is expected to be at least 50% effective.”

Reviewer 2 Report

The concept of the work is simple. The design of experiment is simple too. It is hard to see the novelty of the work. Many errors of presentation are remaining in the manuscript. Therefore, the present manuscript could not be recommended to publish in the Membranes.

Author Response

Reviewer 2

Thank you very much for your revision and your considerations about the paper. We hope the content of this manuscript has been improved after the revision with your comments. We will answer each of them.

The concept of the work is simple. The design of experiment is simple too. It is hard to see the novelty of the work. Many errors of presentation are remaining in the manuscript. Therefore, the present manuscript could not be recommended to publish in the Membranes.

The corrections indicated by the other reviewers have been made and the presentation errors have been revised in the hope that these changes will improve the quality of the article. The authors consider the quality of the manuscript has improved.

Reviewer 3 Report

The manuscript reports a study on an algae membrane photobioreactor for urban wastewater regeneration. There are some significant flaws in the manuscript as listed below.

Major points:

1.       Lines 55-56: References 20-21 do not support the sentences. Improper citation is a kind of academic misconduct. Please check all cited references (not only for Ref. 20 and 21) and make sure that their contents can support the citing sentences.

2.       Figures 2 &3: Why are there gaps in these figures? And why are the gaps in different periods? For example, gaps are ~10-20 days and 60-70 days in TSS, 10-20 and 55-65 days in NTU, 5-15, 40-45, and 50-60 days in OD. Are these data measured in three runs or one run?

3.       Introduction: Many advantages are introduced, but no disadvantage/limitation of MPBR is mentioned in the Introduction. An objective description of its disadvantages/limitations is also needed, despite whether the authors/manuscript or not overcome these disadvantages.

4.       Figure 1: More explanation is needed in the legend text. Where is the membrane? LC and LT meaning? The meaning of symbols/components in the figure should be labeled or given in a figure legend.

Minor points:

1.       Abstract: WWTP should be spelled out.

2.       Line 99: What is PVDF? Full spell-out should be given in the first use of abbreviations.

3.       Line 189: What is LHM?

4.       The unit for conductivity should be like uS/cm, not uS.

5.       Figures: Non-standard abbreviations should be spelled out in the figure or legend, without referring to the main text.

6.       Figures 5 & 6: Texts are overlapped in x-axis labels.

7.       Figure 5b, wrong y-axis label.

8.       Line 235: From the figure, it is not clear when the stationary phase is reached.

9.       There are many “Error! Reference source not found” in the manuscript PDF file.

Author Response

The manuscript reports a study on an algae membrane photobioreactor for urban wastewater regeneration. There are some significant flaws in the manuscript as listed below.

Thank you very much for your revision and your considerations about the paper. We hope the content of this manuscript has been improved after the revision with your comments. We will answer each of them.

Major points:

  1. Lines 55-56: References 20-21 do not support the sentences. Improper citation is a kind of academic misconduct. Please check all cited references (not only for Ref. 20 and 21) and make sure that their contents can support the citing sentences.

Answer: Thank you so much for your recommendation. References cited in the manuscript have been checked to ensure that their content supports the sentences in which they are cited.

  1. Figures 2 &3: Why are there gaps in these figures? And why are the gaps in different periods? For example, gaps are ~10-20 days and 60-70 days in TSS, 10-20 and 55-65 days in NTU, 5-15, 40-45, and 50-60 days in OD. Are these data measured in three runs or one run?

Answer:  Sampling despite the plant operating continuously sampling is carried out in a timely manner since the methods of analysis need to collect a sample and all want to be done at the same time significant of the process.

  1. Introduction: Many advantages are introduced, but no disadvantage/limitation of MPBR is mentioned in the Introduction. An objective description of its disadvantages/limitations is also needed, despite whether the authors/manuscript or not overcome these disadvantages.

Answer: Thank you so much for your recommendation. The following text has been added: “However, a potential drawback of MPBRs is that if the effluent is untreated wastewater, it can lead to the death of the microalgae species being cultivated in the effluent, in which case it is necessary to design an appropriate pre-treatment. In the same way, it is important to correctly select the species of microalgae to be cultivated, because not all of them are capable of adapting to the conditions of the wastewater [35]. Another disadvantage of these photobioreactors is the high risk of contamination of the microalgae culture, as well as the tedious and costly work involved in harvesting the microalgae [36].”

  1. Figure 1: More explanation is needed in the legend text. Where is the membrane? LC and LT meaning? The meaning of symbols/components in the figure should be labeled or given in a figure legend.

Answer: Thank you so much for your suggestion. The legend has been included in the figure, specifying the meaning of the symbols or components present in it.

Minor points:

  1. Abstract: WWTP should be spelled out.

Answer: Thank you so much for your recommendation. The meaning of the abbreviation WWTP has been included in the text.

  1. Line 99: What is PVDF? Full spell-out should be given in the first use of abbreviations.

Answer: Thank you so much for your recommendation. The meaning of the abbreviation PVDF has been included in the text.

  1. Line 189: What is LHM?

Answer: Thank you very much for your consideration. The acronym LHM refers to the units of permeate flow (litres per square metre per hour filtered, L m-1 h-1). This clarification has been introduced in the manuscript.

  1. The unit for conductivity should be like uS/cm, not uS.

Answer: Thank you very much for your consideration. The error in the conductivity units has been corrected.

  1. Figures: Non-standard abbreviations should be spelled out in the figure or legend, without referring to the main text.

Answer: Thank you so much for your recommendation. These abbreviations have been revised and specified in the figures for correct understanding.

  1. Figures 5 & 6: Texts are overlapped in x-axis labels.

Answer: Thank you very much for your consideration. The error in the Figures 5 and 5 have been corrected.

  1. Figure 5b, wrong y-axis label.

Answer: Thank you very much for your consideration. The error in the y-axis in Figure 5b has been corrected by replacing SST by TSS.

  1. Line 235: From the figure, it is not clear when the stationary phase is reached.

Answer: Thank you so much for your recommendation. Figure 2 does not show the stationary state as the study has focused on the growth phase of the microalgae. Therefore, the sentence has been modified as follows for a better understanding: “In this case the results obtained during the growth phase have been represented up to its maximum point, where the stationary phase, not represented, would begin.”

  1. There are many “Error! Reference source not found” in the manuscript PDF file.

Answer: Thank you very much for your consideration. All these reference errors have been corrected in the manuscript.

Reviewer 4 Report

1. The English used in the abstract, main text and conclusion sections of this article has to be improved significantly.

2. In Figure 1, authors should specify what is LT and LC (level transmitter and level control) and the symbol used with a play button has to be explained.

3. Line 154, Error! Reference source not found.  has to be replaced with Table 1. This mistake is repeated throughout the article and has to be corrected. I had very hard time connecting the text to the corresponding tables and figures while reading the manuscript.

4. There are two pictures in Figure 4, they have to be labelled (a) and (b) and further have to be cross-referred in the main text with explanation.

5. In Figures 5 and 6, on the y-axis, LN has to be replace with ln to represent the natural logarithm

6. Lines 312 and 313, authors have to clearly explain how Ri'  and RF' are calculated as it is not clear from the explanation provided in these sentences.

7. The R2 values reported in Tables 4, 5, and 6 correspond to which fit? Is this for the line passing through the origin in Figure 7? Authors need to explain this properly.

8. Line 453, “Respect the membrane” - does it mean “with respect to the membrane”?

9. Conclusion section needs to be improved with good English. I suggest to remove the numbering (1, 2, 3 ,4) within this section and make it as a single paragraph.

Author Response

Reviewer 4

Thank you very much for your revision and your considerations about the paper. We hope the content of this manuscript has been improved after the revision with your comments. We will answer each of them.

  1. The English used in the abstract, main text and conclusion sections of this article has to be improved significantly.

Answer: Thank you so much for your recommendation. The manuscript has been revised and improved by a professional Prof Reading Service including the revision in the present manuscript version. The certificate of the English grammar revision is attached.  

  1. In Figure 1, authors should specify what is LT and LC (level transmitter and level control) and the symbol used with a play button has to be explained.

Answer: Thank you so much for your suggestion. The legend has been included in the figure, specifying the meaning of the symbols or components present in it.

  1. Line 154, Error! Reference source not found.  has to be replaced with Table 1. This mistake is repeated throughout the article and has to be corrected. I had very hard time connecting the text to the corresponding tables and figures while reading the manuscript.

Answer: Thank you very much for your consideration. All these reference errors have been corrected in the manuscript.

  1. There are two pictures in Figure 4, they have to be labelled (a) and (b) and further have to be cross-referred in the main text with explanation.

Answer: Thank you so much for your recommendation. Figures 4a and 4b have been labelled and referenced in the text, pointing out that they show microscope photographs showing the microalgae species present in the photobioreactor.

  1. In Figures 5 and 6, on the y-axis, LN has to be replace with ln to represent the natural logarithm.

Answer: Thank you very much for your consideration. All these reference errors have been corrected in the manuscript. LN has been replaced by ln ln to represent the natural logarithm in Figures 5 and 6.

  1. Lines 312 and 313, authors have to clearly explain how Riand RF'are calculated as it is not clear from the explanation provided in these sentences.

Answer: Thank you so much for your recommendation. The following text has been introduced to explain how the calculations have been made: “For this purpose, the permeate flux and transmembrane pressure results were processed to obtain the membrane permeability (k) in each case. Knowing the relationship of this parameter with the membrane resistance (Equation 2), Ri’ was obtained for each of the tests carried out. The RM’ has been obtained in the same way using the data of the clean membrane. Subsequently, and taking into account Equation 3, the RF’ has been obtained in each case.”

  1. The R2values reported in Tables 4, 5, and 6 correspond to which fit? Is this for the line passing through the origin in Figure 7? Authors need to explain this properly.

Answer: Thank you so much for your recommendation. To explain what is meant by the correlation rate included in Tables 4, 5 and 6, the following sentence has been introduced in the text: "The correlation rate, in each of the cases, corresponds to the data fit by forcing the line through the origin".

  1. Line 453, “Respect the membrane” - does it mean “with respect to the membrane”?

Answer: Thank you so much for your consideration. The error has been reviewed and corrected.

  1. Conclusion section needs to be improved with good English. I suggest to remove the numbering (1, 2, 3 ,4) within this section and make it as a single paragraph.

Answer: Thank you so much for your recommendation. The manuscript has been revised and improved with good English by a professional ProfReadingService. The conclusion has been rewritten in the following way:

“The following conclusions were reached after 80 days under continuous operation (two days of HRT) of a microalgae membrane photobioreactor for the treatment of real urban wastewater effluent by culturing the microalgae present in it.

On one hand, by analysing the results obtained, it was observed that optical density was the parameter that best described the growth kinetics of microalgae present in the photobioreactor. The zero-order model had the highest correlation rate of 0.9776 and a value of 8.921·10-3 days-1 was obtained for the kinetic constant k.

On the other hand, with respect to the membrane, there was an increase in the total membrane resistance of approximately 22%. So under the operative conditions tested, the fouling was low. A two-phase cleaning protocol (inorganic and organic cleaning) was proposed to recover membrane permeability. The recoveries were approximately 42 and 58% in the inorganic and organic phases, respectively, for the temperatures tested (20, 25 and 30°C), but RM’ values were lower as the temperature increased, as would be expected. At all temperatures, the overall effectiveness of the cleaning protocol was 100%. However, the optimum temperature tested for the cleaning protocol is considered to be 30ºC, where a lower RM’ value is obtained.

Regarding the effluent obtained and the removal of the nutrients present in the treated wastewater, total nitrogen and phosphorus removals in the algae membrane photobioreactor were 56.3 and 64.27%, respectively. Therefore, and taking into account other water quality parameters, the effluent of the microalgae membrane photobioreactor could be reused for agricultural irrigation, according to Regulation (EU) 2020/741 of 25 May 2020.”

Round 2

Reviewer 1 Report

Recommended for consideration for publication.

Reviewer 2 Report

The manuscript is revised. Even though, it did not reflect fully the main content related to the “Nutrient removal and membrane performance of an algae membrane photobioreactor in urban wastewater regeneration”. Results on the 3.1 Photobioreactor operation; 3.2.1 Membrane characterization; 3.2.2 Evaluation of the effectiveness of the cleaning protocol and recovery of membrane characteristics; … are not related to the topics. They also do not support to the objectives. Morrover, simple results on “Analysis of the operational TMP” and 3.3 Effluent quality” are presented. Very simple results on “3.3.1 Nitrogen and phosphorus removal” are presented and discussed too.

Again, the concept of the work is simple. The design of experiment is simple too. It is hard to see the novelty of the work. Therefore, the manuscript could not be recommended to publish in the Membranes.

Reviewer 3 Report

I have no further comment.